# Effect of Positive Mental Health on Elderly Patients with Chronic Diseases: The Chain-Mediated Effects of Gratitude and Forgiveness Tendencies at a Tertiary Hospital in Guangzhou

**DOI:** 10.3390/healthcare13050444

**Published:** 2025-02-20

**Authors:** Hai-Cheng Liu, Ying Zhou, Chun-Qin Liu, Xiao-Bing Wu, Graeme D. Smith, Thomas Kwok-Shing Wong, Xin-Yang Hu, Yi-Meng Liu, Ying-Ying Qin, Wen-Jing Wang

**Affiliations:** 1School of Nursing, Guangzhou Medical University, Guangzhou 511436, China; liuhaicheng0210@163.com (H.-C.L.); liuchunqin1990@163.com (C.-Q.L.);; 2Health Medical School, Guangzhou Huashang College, Guangzhou 511300, China; 3The First Affiliated Hospital of Guangzhou Medical University, Guangzhou 510120, China; 4School of Health Sciences, Saint Francis University, Hong Kong SAR, China

**Keywords:** elderly patients with chronic diseases, mature happiness, positive mental health, gratitude, forgiveness tendency, chain mediation

## Abstract

**Background:** This study aims to elucidate the mediating roles of gratitude and forgiveness tendency between mature happiness and positive mental health, seeking to reveal the significance of enhancing these psychological traits to enhance the well-being of this population, so as to provide a theoretical foundation for strategies aimed at promoting healthy aging and enhancing the happiness of elderly patients with chronic illness. **Methods:** From April to October 2024, elderly patients with chronic diseases in the geriatric department of a tertiary hospital in Guangzhou China were selected as subjects through convenience sampling. The study utilized the general information questionnaire, the Chinese Version of Mature Happiness Scale-Revised, the Positive Mental Health Scale, the Gratitude Questionnaire-6, and the Tendency to Forgive Scale for data collection. **Results:** A total of 319 valid questionnaires were obtained. The mature happiness, positive mental health, gratitude, and forgiveness tendency of elderly patients with chronic diseases were at a medium level. Correlation analysis indicated that mature happiness was positively associated with positive mental health, gratitude, and forgiveness tendency. Mediation analysis illuminated that positive mental health robustly predicted the extent of gratitude, which served as a pivotal predictor of mature happiness. Moreover, gratitude and forgiveness tendency together played a significant chain mediating role between mature happiness and positive mental health, accounting for 26.31% of the total effect (*p* = 0.246). **Conclusions:** Positive mental health has a significant positive impact on mature happiness amongst elderly patients with chronic diseases, mediated through the chain mediating effects of gratitude and forgiveness tendency. Consequently, fostering traits of gratitude and forgiveness in this population may enhance their mental health and mature happiness.

## 1. Introduction

Globally, the continuous growth of the elderly population has shifted the focus of public health towards promoting healthy aging. As life expectancy increases, the prevalence and burden of chronic diseases have escalated [1]. Chronic diseases are characterized by their repetitive nature, long duration, poor prognosis, numerous sequelae, and complications, and the associated economic and social burdens significantly impact the physical and mental health of the elderly [2]. Healthy aging encompasses not only physical health but also psychological well-being and happiness [3]. However, traditional psychological research on elderly patients with chronic diseases primarily emphasizes negative emotions such as depression, anxiety, and loneliness, often overlooking the crucial role of internal positive qualities and emotions in maintaining overall health. Studies indicate that enhancing well-being can help delay the progression of chronic diseases, alleviate negative emotions, and promote patient recovery [4].

In traditional positive psychology, happiness is frequently characterized as a fleeting and variable experience centered on positive emotions like satisfaction and pleasure [5]. Nonetheless, this definition possesses limitations. It rests upon a simplistic binary perspective that bifurcates emotions into positive and negative, disregarding the notion that the essence of happiness may lie in a dynamic equilibrium between these opposing emotional states, rather than the supremacy of a single emotion type. To a certain extent, this outlook inadequately accounts for individuals’ internal stability and adaptability when confronted with life’s complexities and uncertainties, nor does it proficiently elucidate how individuals attain enduring happiness amidst multifaceted and volatile life circumstances [6,7].

With the advancement of Existential Positive Psychology (EPP) [8], scholars have delved into the mechanisms by which individuals sustain enduring happiness despite facing adversities. EPP emphasizes the positive aspects of life and seeks a balance between negative and positive elements, moving beyond a simple binary view in order to cultivate a more flexible and mature happiness [9,10]. Compared to the traditional definition of happiness, mature happiness is a more persistent and profound state of well-being that focuses on the individual’s ability to maintain a positive attitude in the face of challenges [11]. It encompasses traits such as inner harmony, self-acceptance, and life satisfaction [12]. Furthermore, mature happiness stems from positive cognition and evaluations of oneself, others, and life, developed through continuous self-growth and adaptation over time [13]. Studies indicate that mature happiness is a stronger predictor of psychological distress, such as stress, anxiety, and loneliness, in comparison to traditional subjective happiness measures [14]. Consequently, investigating the internal mechanisms underlying well-being, particularly mature happiness, holds significant practical implications for elderly patients with chronic diseases.

Positive mental health serves as a fundamental pillar of psychological well-being and it can help to motivate individuals to pursue self-improvement in response to environmental changes [15], playing a crucial role in enhancing overall well-being and reducing anxiety and loneliness amongst elderly [16]. Gratitude, defined as the ability to perceive and appreciate positive aspects of life, encourages individuals to focus on the bright side of life, thereby effectively preventing the emergence of negative emotions [17]. Additionally, forgiveness tendency is an intrinsic and stable personality characteristic that enables individuals to transition from negative to positive states when confronted with interpersonal offenses or adverse situations [18].

Evidence has demonstrated that the psychological traits of gratitude and forgiveness can have a significant positive impact on individuals’ mental health and quality of life. Specifically, gratitude can enhance an individual’s positive emotions and life satisfaction [19], whilst forgiving others can help to reduce levels of psychological stress and negative emotions [20]. For patients with chronic illnesses, these psychological traits can facilitate mental health status improvement [21]. García-Vázquez [22] further revealed a positive correlation between tendencies toward gratitude and forgiveness, individuals with higher levels of gratitude are more likely to exhibit prosocial behaviors and be more inclined to forgive others. This combination of psychological traits plays a positive role in promoting individuals’ mental health and social interactions. Levine [23] noted that chronic disease patients often face physical and emotional fatigue, leading to increased negative emotions and decreased mental health, thereby lowering their overall happiness levels. However, high levels of gratitude can effectively mitigate the negative effects caused by chronic diseases. Additionally, Chen [24] demonstrated that with social and family support, individuals can explore the world through a grateful perspective, actively engage in healthy behaviors, and manage their health more effectively, thereby slowing disease progression and extending life expectancy. Furthermore, studies by Vosvick [25] and Charzyńska [26] indicated that when faced with poor health conditions, self-forgiveness, gratitude, and forgiveness towards others can provide psychological capital to improve patient well-being, enhancing their psychological resilience and adaptive capacity. According to PERMA theory [27] and the cognitive adaptation theory [28], positive mental health, gratitude, and forgiveness tendency act as critical psychological capital, enhancing positive emotions and facilitating cognitive remodeling. This process can enable patients to reframe chronic disease related challenges, promoting a sense of life engagement and meaning, which is crucial for improving mature happiness in elderly patients with chronic diseases.

However, to date, limited research exists on the correlation between mature happiness and positive mental health, gratitude, and forgiveness tendency in global, particularly amongst elderly patients with chronic diseases. This study aims to investigate these relationships to enhance the understanding of how positive mental states collaboratively promote mature happiness. By doing so, this research seeks to enrich the theoretical frameworks of positive psychology and geriatric psychology. These findings may provide insights into methods and strategies for enhancing the well-being of elderly patients with chronic diseases, thereby providing theoretical support for active aging. This study posits the following hypotheses, which are illustrated in Figure 1:
(1)Demographic and disease-related factors may influence mature happiness in elderly patients with chronic diseases.(2)Positive mental health, gratitude, and forgiveness tendency are significantly associated with mature happiness in elderly patients with chronic diseases.(3)Positive mental health will influence mature happiness through the mediating effect of gratitude or forgiveness tendency.(4)Gratitude and forgiveness tendency play a chain mediating role in the influence of positive mental health and mature happiness.

## 2. Methods

### 2.1. Study Design

The study adhered to the STROBE guidelines for cross-sectional research and employed a descriptive cross-sectional design.

### 2.2. Participants

From May to September 2024, a convenience sampling method was employed to recruit elderly patients with chronic diseases from the geriatric department of a tertiary hospital in Guangzhou China for a questionnaire survey. Inclusion criteria: (1) Age ≥ 60 years old [29]; (2) Diagnosis of at least one chronic disease as defined by the International Classification of Diseases, 11th Revision (ICD-11) [30], including but not limited to chronic obstructive pulmonary disease, hypertension, and diabetes; (3) Normal language expression and comprehension abilities; (4) Provided informed consent and voluntarily participated in the study. Exclusion criteria: (1) Conditions indicative of critical health status or terminal phase, including but not limited to organ failure and severe traumatic injuries; (2) Severe mental illness or cognitive impairment precluding normal communication; (3) Suffering major changes within half a year.

### 2.3. Sample Size

Based on the sample size estimation method for quantitative studies [31], the required sample size was calculated to be 5–10 times the number of independent variables. Considering a potential 20% rate of invalid responses, the target sample size was determined to be between 163 and 325 participants. This study ultimately included 335 questionnaires, thereby meeting the sample size requirement.

### 2.4. Measuring Instruments

#### 2.4.1. Demographics

A self-developed general information questionnaire was employed, encompassing demographic details including gender and age, as well as the status of chronic diseases.

#### 2.4.2. The Chinese Version of the Mature Happiness Scale–Revised (MHS-R-CV)

The Mature Happiness Scale was originally developed by Wong [12], later revised by Carreno [14]. Our research team translated, sinicized, and cross-culturally adapted of the revised English version to create the MHS-R-CV. With a Cronbach’s α coefficient or 0.921, the MHS-R-CV demonstrates strong internal consistency and is well-suited for assessing mature happiness among Chinese residents. The scale comprises seven items that measure aspects such as inner harmony, calmness, acceptance, satisfaction, and overall life satisfaction using a 5-point Likert scale ranging from 1 (not at all) to 5 (all of the time). Total scores range from 7 to 35, with higher scores indicating greater mature happiness.

#### 2.4.3. Positive Mental Health Scale (PMHS)

Originally developed by German scholars Lukat [32], the PMHS aims to assess the internal and external factors of individual positive mental health and consists of 9 items. It uses a Likert 4-point rating scale, with higher scores indicating higher levels of positive mental health. The scale has been validated and applied in several countries, including France, Germany, and Russia [33]. The Chinese adaptation of the PMHS was employed in this study, demonstrating a Cronbach’s α coefficient of 0.95 and a test–retest reliability of 0.69, indicating satisfactory psychometric properties within the Chinese population [34].

#### 2.4.4. The Gratitude Questionnaire-6 (GQ-6)

The Gratitude Questionnaire-6 was developed by Emmons and McCullough [35]. It is used to evaluate the frequency and intensity of individuals’ feelings of gratitude, as well as the density and breadth of events that produce gratitude, so as to reflect individual differences in the characteristics of gratitude. The scale contains six affective, thought, or behavioral items related to gratitude. In our study, the Cronbach’s α coefficient of the Chinese version of the scale was 0.81 [36]. The Likert 7-point scoring method was used, in which 1 represented “strongly disagree” and 7 represented “strongly agree”. Items 3 and 6 needed to be scored in reverse. The total score spans from 6 to 42, with elevated scores signifying greater levels of gratitude traits in the individual.

#### 2.4.5. Tendency to Forgiveness Scale (TTF)

This scale was developed by Brown [37] to assess the level of an individual’s tendency to forgive from the perspective of temperament. The scale contains 4 items and is scored on a 5-point Likert scale ranging from 1 “strongly disagree” to 5 “strongly agree”, with higher scores indicating a greater tendency to forgive. The scale showed good internal consistency reliability (α = 0.82) and test–retest reliability (r = 0.71). The Chinese version also shows good reliability (α = 0.61) and validity, and has been widely used for studies involving college students [38].

### 2.5. Research Methodology

In this survey design study, two professionally trained nursing graduate students served as investigators, and data were collected using structured questionnaires and face-to-face interviews. Before filling in the questionnaire, the investigator explained the purpose and significance of the study in detail to the elderly patients with chronic diseases who participated in the study, filled it in in an anonymous manner after obtaining the patient’s consent, and withdrew it immediately after completion. If the study participant had poor vision or low education, the investigator assisted in filling out the questionnaire based on the patient’s verbal responses. A total of 335 questionnaires were collected in this survey, including 16 invalid questionnaires and 319 valid questionnaires, with an effective recovery rate of 95.22%.

### 2.6. Statistical Analysis

SPSS 27.0 software was used for data analysis. Count data were described as frequency and percentage. The measurement data conforming to normal distribution were expressed as mean (M) and standard deviation (SD). Two independent sample t test, or analysis of variance was used for comparison between groups. Pearson correlation analysis was used to explore the correlation between mature happiness and positive mental health, gratitude and forgiveness tendency in elderly patients with chronic diseases. SPSS macro PROCESS 4.1 was used to test the mediation effect.

## 3. Results

### 3.1. Common Method Bias Test

In this study, Harman single factor was used to detect common method bias. The results showed that a total of four factors with eigenvalues greater than 1 were extracted, and the first factor explained 36.43% of the variance, which was lower than the critical standard of 40% [39]. Controlling for the effects of an unmeasured latent methods factor (ULMC) also corroborated this assertion. The comparison between the bifactor model (RMSEA = 0.062, CFI = 0.916, TLI = 0.906) and the model with only trait factors (RMSEA = 0.069, CFI = 0.893, TLI = 0.881) revealed no significant differences. The improvement in fit indices was minimal, with increases in CFI and TLI values less than 0.1 and a decrease in RMSEA value less than 0.05 [40]. Therefore, the common method bias in this study can be considered not to be a serious problem.

### 3.2. Sample Characteristics

This study included 319 elderly patients with chronic diseases aged between 60 and 91 years, with an average age of (68.33 ± 6.76) years. The highest proportion of patients was aged between 60 and 70 years (70.2%). There were 160 male patients (50.2%) and 159 female patients (49.8%). The majority of study participants were married (84.6%), and half (49.8%) lived with their spouses. The majority of patients had a monthly pension income of between CNY 2000 and 3999 (43.9%). One hundred and forty-two patients had a chronic disease history of over 10 years or more, and most patients took more than two medications daily (65.2%). More than half of the patients had two or more chronic diseases (66.8%). The univariate analysis and *t* test found that there were statistically significant differences in the MHS-R-CV scores of elderly patients with chronic diseases in terms of the three variables of current living style, marital status, and perceived health status (*p* < 0.05). Refer to Table 1 for details.

### 3.3. Descriptive Statistic and Correlation Analysis Among Variables

The scores of MHS-R-CV, PMHS, GQ-6, and TFF of 319 elderly patients with chronic diseases were (21.02 ± 6.77), (22.91 ± 5.01), (25.05 ± 5.61), and (13.14 ± 2.95), respectively. The results of correlation analysis showed that the mature happiness of elderly patients with chronic diseases was positively correlated with positive mental health, gratitude, and forgiveness tendency (r = 0.691, 0.612, 0.590, all *p* < 0.01), and positive mental health was positively correlated with gratitude and forgiveness tendency (r = 0.404, 0.373, all *p* < 0.01). Gratitude was positively correlated with forgiveness tendency (r = 0.729, *p* < 0.01). See Table 2.

### 3.4. Multiple Linear Regression Analysis of Influencing Factors of Mature Happiness in Elderly Patients with Chronic Diseases

Taking mature happiness as the dependent variable, three statistically significant variables in the univariate analysis and *t* test (*p* < 0.05), current living style, marital status and perceived health status, were used as control variables, and a silent variable was set. In addition, positive mental health, gratitude, and forgiveness tendency were used as independent variables, and multiple linear regression analysis was conducted by stepwise method. The independent variable assignment is shown in Table 3.

Through stepwise regression analysis, a total of three models were obtained. Ultimately, the variables of positive mental health, gratitude, and forgiveness tendency were included in the model, indicating that these three variables are the influencing factors of mature happiness of elderly chronic disease patients. The coefficient of determination R^2^ of the regression model was 0.633, adjusted R^2^ was 0.630, F was 181.105 (*p* < 0.001), suggesting that the model fits well, and the variables entered into the model can explain 63.0% of the total variance of mature happiness of elderly chronic disease patients. In addition, the tolerance value in this study was 0.449 to 0.823, and the variance inflation factor (VIF) was 1.215 to 2.228 < 10 [41], indicating that there is no serious multicollinearity problem among the variables, and the detailed results are shown in Table 4.

### 3.5. Analysis of the Chain Mediation Effect Between Mature Happiness and Positive Mental Health in Elderly Patients with Chronic Diseases

In order to deeply explore the positive effects of positive mental health, gratitude and forgiveness tendency on mature happiness and its internal mechanism, this study took positive mental health as the dependent variable (*X*), gratitude and forgiveness tendency as the mediating variable (*M*), and mature happiness as the independent variable (*Y*). Multivariate hierarchical regression analysis under Model 6 was performed using the PROCESS 4.1 macro program of SPSS software. The results showed that positive mental health had a significant positive predictive effect on gratitude and mature happiness (β = 0.453, 0.689, both *p* < 0.001). Positive mental health had a positive predictive effect on forgiveness tendency (β = 0.055, *p* = 0.025). In addition, gratitude showed a significant positive predictive effect on forgiveness tendency and mature well-being (β = 0.363, 0.295, both *p* < 0.001), and forgiveness tendency also had a significant positive predictive effect on mature happiness (β = 0.510, *p* < 0.001) (see Table 5).

Gratitude and forgiveness tendency had a chain mediating effect on positive mental health and mature happiness in elderly patients with chronic diseases, and the mediating effect value was 0.246, accounting for 26.31% of the total effect (0.935). Positive mental health affects mature happiness through the following four paths: (1) direct path: positive mental health → mature happiness; (2) indirect path 1: positive mental health → gratitude → mature happiness; (3) indirect path 2: positive mental health → forgiveness tendency → mature happiness; (4) indirect path 3: positive mental health → gratitude → forgiveness tendency → mature happiness. The relational model is shown in Figure 2.

A total of 5000 bootstrap samples were randomly and repeatedly selected from 319 elderly patients with chronic diseases to test the chain mediating effect of gratitude and forgiveness tendency between positive mental health and mature happiness of elderly patients with chronic diseases. The results showed that the 95% CI of the three indirect paths did not include 0, indicating that the independent mediating effect of gratitude and forgiveness tendency and the chain mediating effect of gratitude and forgiveness tendency were significant. The proportion of the three indirect path effects was 14.33%, 2.99%, and 8.98%, respectively, as detailed in Table 6.

## 4. Discussion

Quality of Life (QoL) is a comprehensive and holistic concept that encompasses an individual’s overall state and satisfaction across multiple dimensions, including physical health, psychological well-being, social relationships, environment, and economic status [42]. For elderly patients with chronic diseases, the long-term impact of illness not only leads to physical decline but also induces feelings of helplessness and despair. Moreover, it frequently gives rise to psychological issues such as loneliness, anxiety, and depression [43]. These psychological challenges further diminish their overall satisfaction and erode their quality of life. Therefore, addressing the psychological issues of elderly patients with chronic diseases is of paramount importance for alleviating the burden of illness and enhancing their quality of life. In this study, we have shown the interactions between positive mental health, gratitude, forgiveness tendencies, and mature happiness in older adults with chronic diseases. Preliminary results showed that the score of elderly patients with chronic diseases in MHS-R-CV was (21.02 ± 6.77) points, slightly higher than the median total score of 21 points, showing a medium level of mature well-being, which still needs to be further improved, similar to the research results of Carreno [14]. This may be due to the long-term existence of chronic diseases and the coexistence of multiple diseases, which make elderly face a higher risk of disability and the threat of a shortened health life span, potentially causing psychological distress, leading to decreased happiness [44]. Therefore, health care professionals need to have awareness of the importance of identifying and screening for mental health problems in elderly patients with chronic diseases as early as possible, to enable mental health interventions and management measures in order to improve their overall well-being.

Further correlational analyses revealed significant positive associations between mature happiness and high levels of positive mental health, gratitude, and forgiveness tendencies in older adults with chronic diseases, similar to previous findings [45,46]. This suggests that elderly patients with chronic diseases who have better positive mental health status and higher levels of gratitude and forgiveness tendency experience stronger mature happiness. This may be explained as those with personality traits often have better emotional stability and self-efficacy, a greater ability to quickly adjust their mindset to relieve the discomfort and stress caused by the disease, leading them to view the disease in a more positive perspective, be more self-focused, and have better recovery beliefs [47]. Secondly, gratitude can stimulate the positive emotional experience of elderly patients with chronic diseases, alleviate the pain caused by excessive attention to the disease, optimize their interpersonal relationships, build a more solid social support system, and thus enhance their mature happiness [48]. Finally, forgiveness tendency helps patients release negative emotions, reduce psychological burden, promote self-acceptance, enhance inner peace and tranquility, and further enhance happiness experience [49].

In exploring the relationship between positive mental health and mature happiness, this study revealed the mediating roles of gratitude and forgiveness tendencies. This finding holds significant implications for the clinical management of elderly patients with chronic illnesses. Mediation analysis indicated that gratitude and forgiveness tendencies significantly mediate the relationship between positive mental health and mature happiness, accounting for 26.31% of the total effect. These results suggest that a positive mental health state can indirectly enhance the level of mature happiness in elderly patients with chronic illnesses by increasing their tendencies towards gratitude and forgiveness.

In clinical practice, this may imply that psychological interventions such as gratitude training, forgiveness promotion activities, or mindfulness meditation can help patients focus more on self-awareness and acceptance, effectively improving their levels of positive mental health and better equipping them to face the challenges posed by chronic diseases [50,51]. These interventions not only improve emotional states but also may help to promote psychological maturity and well-being by enhancing gratitude and forgiveness tendencies, thereby playing an active role in chronic disease management [52]. Therefore, cultivating these traits can enhance patients’ psychological adaptability, increase their life satisfaction and vitality, and reduce negative emotions, such as depression and envy [53]. Additionally, the cultivation of gratitude and forgiveness can be achieved through narrative nursing, a method proven to effectively alleviate depression and anxiety symptoms in chronic disease patients, enhancing their quality of life and treatment adherence [54]. Through narrative nursing, patients are better able to understand their emotions and experiences, achieving psychological growth and maturity [55], which aligns with the concept of mature happiness. By considering these psychological tendencies comprehensively and incorporating them into treatment and care plans, the overall well-being of patients can be effectively enhanced.

### Limitations

The limitation of this study is that only one tertiary hospital in Guangzhou was selected as the survey object, and the sample size was small and may be affected by regional differences. Therefore, it is necessary to expand the scope of investigation, increase the sample size, and construct more targeted and universal intervention programs in the future, so as to better evaluate the mature happiness level of this special population.

## 5. Conclusions

In conclusion, positive mental health not only directly affects mature happiness, but also indirectly promotes mature happiness by enhancing gratitude, and the chain mediating effect of gratitude and forgiveness tendency. Therefore, in the intervention to improve the mature happiness of elderly patients with chronic diseases, gratitude and forgiveness can be cultivated to improve their mature happiness. Future research can further explore the differences in mediating effects of gratitude and forgiveness on mature happiness in the elderly patients with chronic diseases under different cultural backgrounds to verify its universality. At the same time, long-term follow-up studies can be carried out to observe the lasting effects of interventions to cultivate gratitude and forgiveness tendency on mature happiness of the elderly patients with chronic diseases.

## Figures and Tables

**Figure 1 healthcare-13-00444-f001:**
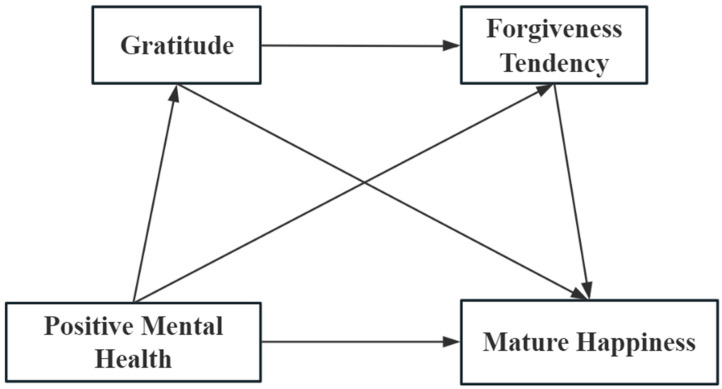
Hypotheses model.

**Figure 2 healthcare-13-00444-f002:**
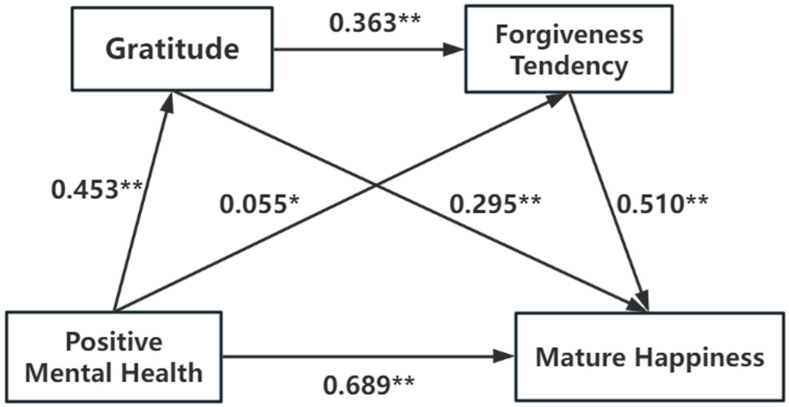
Chain mediation model of gratitude and forgiveness tendency in positive mental health and mature happiness of elderly patients with chronic diseases (Note: ** indicates *p* < 0.001, * indicates *p* < 0.05).

**Table 1 healthcare-13-00444-t001:** Sample characteristics and univariate analysis (*n* = 319).

Variable	*n*	%	MHS-R-CV Score (Mean ± SD)	*t* */F*	*p*
**Gender**				−1.208	0.228
Male	160	50.2	20.56 ± 6.61		
Female	159	49.8	21.48 ± 6.92		
**Age (years)**				2.871	0.058
60–70	224	70.2	20.46 ± 6.80		
71–79	72	22.6	22.01 ± 6.78		
≥80	23	7.2	23.3 ± 5.82		
**Current living style**				3.363	0.036
Solitude	27	8.5	17.89 ± 5.83		
Cohabitation	159	49.8	21.10 ± 6.94		
Live with children or relatives	133	41.7	21.56 ± 6.63		
**Marital status**				−2.330	0.020
Married	270	84.6	21.39 ± 6.88		
Not in marriage	49	15.4	18.96 ± 5.77		
**Degree of education**				0.927	0.449
Primary and below	37	11.6	20.70 ± 7.46		
Middle school	91	28.5	21.95 ± 6.69		
High school/technical secondary school	102	32	21.16 ± 6.57		
Diploma	55	17.2	20.16 ± 6.61		
Bachelor degree or above	34	10.7	19.85 ± 7.10		
**Pension income (CNY/monthly)**				2.422	0.066
<2000	35	11	22.14 ± 7.15		
2000–3999	140	43.9	19.90 ± 6.74		
4000–5999	103	32.3	22.03 ± 6.83		
≥6000	41	12.9	21.34 ± 5.99		
**Perceived health status**				6.077	0.003
Good	140	43.9	22.28 ± 6.43		
General	130	40.8	20.58 ± 6.69		
Bad	49	15.4	18.57 ± 7.27		
**Self-care ability of daily living**				0.055	0.956
Yes	308	96.6	21.02 ± 6.80		
No	11	3.4	20.91 ± 6.40		
**Do you exercise every week?**				−0.583	0.560
Yes	89	27.9	20.66 ± 7.09		
No	230	72.1	21.16 ± 6.66		
**Duration of chronic disease**				0.752	0.472
<5 years	65	20.4	21.69 ± 6.98		
5–10 years	112	35.1	20.45 ± 6.88		
>10 years	142	44.5	21.16 ± 6.60		
**Number of daily doses**				1.655	0.099
≤1 type	110	34.5	21.87 ± 6.58		
≥2 types	208	65.2	20.55 ± 6.86		
**Number of hospitalizations in the past year**				1.125	0.261
≤1 time	241	75.5	21.26 ± 6.75		
≥2 times	78	24.5	20.27 ± 6.83		
**Number of chronic diseases**				1.957	0.051
1 type	106	33.2	22.07 ± 6.15		
≥2 types	213	66.8	20.5 ± 7.02		

Note: n = frequency, % = percentage, Mean = arithmetic mean, SD = standard deviation.

**Table 2 healthcare-13-00444-t002:** Pearson correlation coefficient (*n* = 319).

Scale	Score Range	Mean ± SD	MHS-R-CV	PMHS	GQ-6	TFF
MHS-R-CV	10–33	21.02 ± 6.77	1.000			
PMHS	10–36	22.91 ± 5.01	0.691 **	1.000		
GQ-6	14–38	25.05 ± 5.61	0.612 **	0.404 **	1.000	
TFF	6–19	13.14 ± 2.95	0.590 **	0.373 **	0.729 **	1.000

Note: Mean = arithmetic mean, SD = standard deviation. ** indicates *p* < 0.01.

**Table 3 healthcare-13-00444-t003:** Independent variable assignment description.

Variable	Assignment Condition
Current living style	1 = “solitude”; 2 = “cohabitation”; 3 = “live with children or relatives”;
Marital status	1 = “married”; 2 = “not in marriage”
Perceived health status	1 = “good”; 2 = “general”; 3 = “bad”;
Positive mental health	continuous variable
Gratitude	continuous variable
Forgiveness tendency	continuous variable

**Table 4 healthcare-13-00444-t004:** Multiple linear regression analysis (*n* = 319).

Model	Regression Coefficient	SE	Standard Regression Coefficient	*t*	*p*	Tolerance	VIF
Quantity	−8.860	1.308	/	−6.773	<0.001	/	/
Positive mental health	0.689	0.051	0.510	13.546	<0.001	0.823	1.215
Gratitude	0.295	0.062	0.244	4.798	<0.001	0.449	2.228
Forgiveness tendency	0.510	0.115	0.222	4.417	<0.001	0.462	2.166

Note: SE = standard error, VIF = variance inflation factor.

**Table 5 healthcare-13-00444-t005:** Mediation effect of gratitude and forgiveness tendency on positive mental health and mature happiness in elderly patients with chronic diseases.

Regression Equation		Overall Fit Index			Regression Coefficient	
Outcome Variable (*Y*)	Predictive Variable (*X*)	*R*	*R* ^2^	*F*	β	*t*
Gratitude	Positive mental health	0.404	0.164	61.998 **	0.453	7.874 **
Forgiveness tendency	Positive mental health	0.734	0.538	184.203 **	0.055	2.251 *
	Gratitude				0.363	16.522 **
Mature happiness	Positive mental health	0.796	0.633	181.105 **	0.689	13.546 **
	Gratitude				0.295	4.798 **
	Forgiveness tendency				0.510	4.417 **

Note: ** indicates *p* < 0.001, * indicates *p* < 0.05.

**Table 6 healthcare-13-00444-t006:** Total effect, direct effect, and intermediate effect results.

Influence Path	Effect Size	SE	95%CI		Effect Ratio (%)
			Lower	Upper	
Total effect	0.935	0.055	0.827	1.043	/
Direct effect	0.689	0.051	0.589	0.789	73.69
Intermediate effect	0.246	0.032	0.184	0.310	26.31
Positive mental health → Gratitude → Mature happiness	0.134	0.030	0.073	0.192	14.33
Positive mental health → Forgiveness tendency → Mature happiness	0.028	0.014	0.003	0.057	2.99
Positive mental health → Gratitude → Forgiveness tendency → Mature happiness	0.084	0.021	0.045	0.128	8.98

Note: SE = standard error.

## Data Availability

The data are available upon request from the corresponding author.

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
