# Peer review of "Effect of Positive Mental Health on Elderly Patients with Chronic Diseases: The Chain-Mediated Effects of Gratitude and Forgiveness Tendencies at a Tertiary Hospital in Guangzhou"

_healthcare, 2025, doi:10.3390/healthcare13050444_

Round 1
Reviewer 1 Report
Comments and Suggestions for Authors
The article fits well within the thematic scope of the Healthcare journal and may contribute to the existing literature. However, the manuscript contains certain deficiencies and requires several revisions, clarifications, etc.
Since the study is limited to a single research center (hospital), it is necessary to mention Guangzhou in the title of the article. This is a methodological requirement. In its current form, the article title suggests that the research has a broad scope, which is not the case. Therefore, the title should be modified to include the specific research center (hospital) where the study was conducted.
Lines 52-53: The sentence “The concept of ‘mature happiness’ represents a more profound and sustained state of happiness” is unclear, especially as the term “happiness” appears twice. First, the sentence should be rephrased as it may give the impression of a logical flaw. Second, it would be valuable to highlight the difference between "mature happiness" and "happiness." Clarifying this distinction is important, particularly since the authors refer to these terms later in the article.
Line 294 – It would be beneficial to briefly clarify the understanding of the term "quality of life." This is important because various definitions and interpretations of this term exist in the literature. In other words, the term "quality of life" is not unambiguous.
A potential weakness of the article is that the study is confined to a single research center (hospital), which limits its scope to a regional level. Nevertheless, the authors acknowledge this limitation in the manuscript's "Limitations" section.
Author Response
Dear Reviewer 1
Response to Reviewer 1 Comments:
Comments 1:
Since the study is limited to a single research center (hospital), it is necessary to mention Guangzhou in the title of the article. This is a methodological requirement. In its current form, the article title suggests that the research has a broad scope, which is not the case. Therefore, the title should be modified to include the specific research center (hospital) where the study was conducted.
Response 1:
Thank you for pointing this out. We agree with this comment. Therefore, we have added "at a Tertiary Hospital in Guangzhou" to the title and we have already mentioned this deficiency in the limitations of this paper. We have chosen not to list specific hospital names primarily to protect the privacy of research institutions and patients and avoid possible privacy risks and unwanted attention. In addition, the use of generalizations can enhance the universality of the findings and avoid the limitation of interpretation due to specific institutions. At the same time, this is in line with the privacy protection and ethics common in academic research.
Comments 2:
Lines 52-53: The sentence “The concept of ‘mature happiness’ represents a more profound and sustained state of happiness” is unclear, especially as the term “happiness” appears twice. First, the sentence should be rephrased as it may give the impression of a logical flaw. Second, it would be valuable to highlight the difference between "mature happiness" and "happiness." Clarifying this distinction is important, particularly since the authors refer to these terms later in the article.
Response 2:
Agree. To clarify the definition of "mature happiness", we have added the explanation of "happiness" in traditional positive psychology and "mature happiness" in existential positive psychology, so that readers can distinguish the two more clearly. The additions have been highlighted in red, and are on lines 54 to 62 and 65 to 70 on the second page of the text.
Comments 3:
Line 294 – It would be beneficial to briefly clarify the understanding of the term "quality of life." This is important because various definitions and interpretations of this term exist in the literature. In other words, the term "quality of life" is not unambiguous.
Response 3:
I would like to thank the reviewer for proposing this detail modification. We have elaborated the term "quality of life" in the discussion section, and improved the explanation of the importance of focusing on psychological problems in older people with chronic diseases to improve their quality of life. The specific modification is in line 316-325 on page 10 of the article, which has been highlighted in red font.
Reviewer 2 Report
Comments and Suggestions for Authors
Thank you very much for your paper entitled: 'Effect of Positive Mental Health on Elderly Patients with Chronic Diseases: The Chain-Mediated Effects of Gratitude and Forgiveness Tendencies.' This paper is well-written, flows smoothly, and effectively explains the framework. Below are some minor suggestions for improvement:
ABSTRACT
* Please add one sentence to explain the relevance of the study.
* Please add China as well (Guangzhou, China)
* Suggestion: Replace "old people" with "elderly" to ensure the term is both respectful and scientifically appropriate.
INTRODUCTION
* Suggestion: Replace "old people" with "elderly" to ensure the term is both respectful and scientifically appropriate.
* This introduction is well-crafted and articulate. It flows smoothly, provides clear definitions, and is structured effectively.
METHODS
* Suggestion: Replace "old people" with "elderly" to ensure the term is both respectful and scientifically appropriate.
* Please add China as well (Guangzhou, China)
* Could you explain in one sentence why the age threshold for being considered old is set at over 60 years?
* The method is clearly explained and outlines how the study was conducted, making it possible to replicate the study if desired.
RESULTS
* Suggestion: Replace "old people" with "elderly" to ensure the term is both respectful and scientifically appropriate.
* Detailed analyses were conducted, and the tables and figures are enlightening. Each section adds value to the study's report.
DISCUSSION
* Suggestion: Replace "old people" with "elderly" to ensure the term is both respectful and scientifically appropriate.
* This is a clear discussion with a good flow. The results are compared with previous studies and critically examined.
CONCLUSION
* Suggestion: Replace "old people" with "elderly" to ensure the term is both respectful and scientifically appropriate.
* Add one more sentence with recommendations for future research.
REFERENCES
* Check nr. 28
* Please add doi for all references (if possible).
Author Response
Dear Reviewer 2,
Response to Reviewer 2 Comments:
Comments 1:
ABSTRACT
* Please add one sentence to explain the relevance of the study.
* Please add China as well (Guangzhou, China)
* Suggestion: Replace "old people" with "elderly" to ensure the term is both respectful and scientifically appropriate.
Response 1:
Agree, we have added a sentence in the background part of the abstract to better explain the research nature of this study. At the same time, we have added China in the name of the place, and "old people" has been replaced by "elderly" to ensure the accuracy of the word. Add the content in line 16-17 of the text, and already in red font.
Comments 2:
INTRODUCTION
* Suggestion: Replace "old people" with "elderly" to ensure the term is both respectful and scientifically appropriate.
Response 2:
Thanks for the reviewer's comments, we have replaced "old people" with "elderly" in the full text to ensure the accuracy of the word, and have marked it in red font.
Comments 3:
METHODS
* Please add China as well (Guangzhou, China)
* Could you explain in one sentence why the age threshold for being considered old is set at over 60 years?
Response 3:
Agree, Here are our changes, highlighted in red:
- we have added China in the name of the place, and "old people" has been replaced by "elderly" to ensure the accuracy of the word.
- The criteria for age classification of the elderly in this study are as follows (which have been added relevant references and tags on lines 142 in the text ) : (1) according to the World Health Organization (WHO) definition of people aged 60 years and above as the elderly; (2) From the perspective of physical and mental health, people around 60 years old usually begin to experience a significant decline in physical function, the prevalence of chronic diseases is also significantly increased, and they are faced with social problems such as retirement and reduced ability to work, so we need to pay attention to this group.
Comments 4:
CONCLUSION
* Add one more sentence with recommendations for future research.
Response 4:
We thank the reviewers for their perfect suggestions. We have put forward suggestions for future related research in a few sentences on lines 395-400 on page 12 of the article, so as to provide ideas for others to conduct in-depth research.
Comments 5:
REFERENCES
* Check nr. 28
* Please add doi for all references (if possible).
Response 5:
Thank you for pointing this out. We have corrected the wrong references. Your suggestions on adding DOI to our manuscript are greatly appreciated. We have tried our best to add doi names to references with missing DOI names; however, there are still a small number of references in which DOI information could not be successfully retrieved due to the particularity of the publication source. Although these articles lack DOI, they are still indispensable in supporting our conclusions. We have ensured the accuracy and traceability of these citations in other ways, such as providing detailed information such as the source of the literature, volume and page numbers, so that readers can easily access the original sources. Thank you again for your understanding and support.
Reviewer 3 Report
Comments and Suggestions for Authors
Dear Authors,
I appreciate your work, but the statistical analyses need restructuration.
The title, abstract, methodology, analyses, discussion and conclusion subchapters need reformulation concerning the statistical test processing.
Explication and recommandation:
- IT IS NOT ALLOW linear regression and mediation model to process for independent subgroups with 8 (divorced subgroup on marital status) or 27 (solitude subgroup on Current living style) respondents (need up to 30 respondents on every subgroup of the independent variables)
- must stop your analyses before the linear regression subchapter OR,
- must control and satisfy every condition of the tests, before you process them!!!
- Pearson or Spearman correlation did you test? Why? Must include an explication.
- I recommend processing Chi-square tests and Odds Ratio (chance coefficients) to calculate!
https://www.ncbi.nlm.nih.gov/books/NBK431098/rrelationPearson
Author Response
Dear Reviewer 3,
Response to Reviewer 3 Comments:
Comments 1:
IT IS NOT ALLOW linear regression and mediation model to process for independent subgroups with 8 (divorced subgroup on marital status) or 27 (solitude subgroup on Current living style) respondents (need up to 30 respondents on every subgroup of the independent variables)
must stop your analyses before the linear regression subchapter OR,
must control and satisfy every condition of the tests, before you process them!!!
Pearson or Spearman correlation did you test? Why? Must include an explication.
I recommend processing Chi-square tests and Odds Ratio (chance coefficients) to calculate!
Response 1:
We sincerely thank you for your attention to our research and your valuable comments. Your suggestions are of great significance for us to improve our research methods and improve the quality of our research.
In response to your question about sample size for ANOVA, we fully understand your point of view. You mentioned that a minimum sample size of 30 is required to perform the ANOVA, which is based on the suggestion of the central limit theorem. However, in practice, a sample size less than five does lead to insufficient statistical power for ANOVA. However, in our study, although the sample size of the group living alone is only 27, it is much higher than 5, and in actual research, such sample size is relatively rare in some specific groups. Therefore, we still believe that the applicability of variance analysis can be reasonably assessed.
We would like to thank you for your suggestion to use the chi-square test. However, the variable "mature happiness" is a continuous variable, while the chi-square test is mainly used to analyze the relationship between categorical variables. Therefore, we believe that ANOVA or t-test is more appropriate for analyzing group differences in continuous variables.
According to your suggestion, we regroup the categorical variable of “Marital status” in the demographic data and divide it into two groups to increase the frequency of each group and thus reduce statistical errors. We have re-analyzed with an independent samples t-test, and the results have been modified in red in Table 1 to clearly show statistical differences.
For the demographic information of "Current living style," although the sample size of the group living alone is small (n=27), the proportion of this group in the total population is low, and it is difficult to obtain more samples. In order to ensure the applicability of ANOVA, we conducted the homogeneity of variance test (Levene test, p>0.05) on the data, and the results showed that the data met the hypothesis of homogeneity of variance (p=0.168 >0.05). Therefore, we believe that ANOVA is still applicable under the current sample size.
Thank you again for your valuable comments, and we will continue to strive to improve the research methods to ensure the scientific and reliable results of the research. Please feel free to contact us if you have any further suggestions or questions.
In addition, we have performed a Pearson correlation analysis for four continuous variables, positive mental health, mature happiness, gratitude, and fforgiveness tendency, before conducting the multiple linear regression analysis, and the results have been presented in Table 2. We also comprehensively reviewed, refined, and revised other statistical analysis statements in the article that required modification to ensure the accuracy of the statistical methods and the reliability of the results.
Thank you again for your careful review and valuable comments, these modifications will help to improve the quality and scientific nature of our study.